# IFITM1 and IFITM3 Proteins Inhibit the Infectivity of Progeny HIV-1 without Disrupting Envelope Glycoprotein Clusters

**DOI:** 10.3390/v15122390

**Published:** 2023-12-07

**Authors:** Smita Verma, Yen-Cheng Chen, Mariana Marin, Scott E. Gillespie, Gregory B. Melikyan

**Affiliations:** 1Department of Pediatrics, Emory University School of Medicine, Atlanta, GA 30322, USA; smita.verma@emory.edu (S.V.); ychen85530@gmail.com (Y.-C.C.); mmarin@emory.edu (M.M.);; 2Children’s Hospital of Atlanta, Atlanta, GA 30322, USA

**Keywords:** IFITM, HIV-1 Env, dSTORM, envelope glycoprotein clustering

## Abstract

Human interferon-induced transmembrane (IFITM) proteins inhibit the fusion of a broad spectrum of enveloped viruses, both when expressed in target cells and when present in infected cells. Upon expression in infected cells, IFITMs incorporate into progeny virions and reduce their infectivity by a poorly understood mechanism. Since only a few envelope glycoproteins (Envs) are present on HIV-1 particles, and Env clustering has been proposed to be essential for optimal infectivity, we asked if IFITM protein incorporation modulates HIV-1 Env clustering. The incorporation of two members of the IFITM family, IFITM1 and IFITM3, into HIV-1 pseudoviruses correlated with a marked reduction of infectivity. Super-resolution imaging of Env distribution on single HIV-1 pseudoviruses did not reveal significant effects of IFITMs on Env clustering. However, IFITM3 reduced the Env processing and incorporation into virions relative to the control and IFITM1-containing viruses. These results show that, in addition to interfering with the Env function, IFITM3 restricts HIV-1 Env cleavage and incorporation into virions. The lack of notable effect of IFITMs on Env clustering supports alternative restriction mechanisms, such as modification of the properties of the viral membrane.

## 1. Introduction

Interferon-induced transmembrane proteins (IFITMs) are a family of small transmembrane proteins that restrict the entry of diverse enveloped viruses and modulate essential cellular processes [1,2,3,4,5,6,7,8,9]. Of the five known human IFITM family members, IFITM1, IFITM2, and IFITM3 inhibit a wide range of enveloped viruses, including clinically important pathogens such as the HIV-1, Ebola, Influenza A, and Dengue viruses [10,11,12]. IFITMs are active against viruses that enter cells through both pH-independent and pH-dependent mechanisms, albeit to a varying degree of efficiency [3,6,13,14]. Accumulating evidence suggests that these proteins block virus entry by making the host cell membranes more rigid and imposing unfavorable membrane curvature that traps viral fusion at a hemifusion stage [15,16,17,18,19,20,21,22]. However, intriguingly, IFITMs have little or no effect on the Murine Leukemia Virus and arenaviruses such as the Lassa virus and lymphocytic choriomeningitis virus [23,24,25]. Furthermore, several studies have reported that IFITMs can promote infection by the human papillomavirus 16 [25], Aichi picornavirus [26], and human coronavirus OC43 [27,28].

In addition to inhibition of viral fusion when expressed in target cells, IFITM proteins expressed in infected cells incorporate into progeny virions from diverse virus families and inhibit their infectivity [11,12,18,20] via a poorly understood mechanism [29]. This reduction of infectivity has been dubbed “negative imprinting” because the loss of infectivity does not strictly correlate with the levels of IFITM incorporation into virions [18,20,29,30]. There is disagreement regarding whether the virus’ sensitivity to “negative imprinting” is linked to viral glycoproteins or the mode of virus assembly [18,29,30,31,32].

The sensitivity of HIV-1 to IFITMs maps to the Env glycoprotein [12,18,33,34,35]. Envs from different HIV-1 strains show differential susceptibility to virion-incorporated and target cell-expressed IFITMs [18,36,37,38] and are thus classified as sensitive or resistant Envs. The interaction between IFITM3 and HIV-1 Env in virus-producing cells can inhibit the incorporation and proteolytic cleavage of the gp160 precursor, leading to reduced levels of active (processed) Env in virions [18,32,37]. However, this effect appears to be cell type-dependent, as IFITM3 expression in CD4+ T-cells does not affect Env incorporation into progeny virions [11]. In addition, IFITM3 incorporation into viral particles has been reported to favor a partially open conformation of sensitive Envs, thus sensitizing the virus to neutralizing antibodies and Env-targeting compounds [30,33].

On average, an HIV-1 particle bears only ∼8–14 Env trimers [39,40,41]. Given the sparsity of Env glycoproteins, it has been proposed that Env clustering on virions is essential for infectivity [42], as these clusters may serve as hotspots for efficient HIV-1 fusion [43,44,45]. Indeed, whereas the Env mobility on immature particles is likely restricted through Env-Gag interactions, super-resolution microscopy studies have detected the formation of Env clusters following HIV-1 maturation [42,46,47].

Our previous super-resolution imaging results suggested that virus-incorporated SERINC5, another host factor that restricts HIV-1 fusion through incorporation into virions, disrupts Env clusters [35]. We therefore asked whether IFITM incorporation can reduce HIV-1 infectivity by disrupting maturation-dependent Env clustering. Given the high degree of IFITM2 and IFITM3 sequence homology, their localization to endosomes and a similar range of restricted viruses [8,48] we focused on the IFITM1 and IFITM3 proteins. The effect of IFITM1 and IFITM3 incorporation on Env glycoprotein clustering was assessed using direct stochastic optical reconstruction microscopy (dSTORM). We find that neither IFITM1 nor IFITM3 significantly disrupts Env clusters. However, expression of IFITM3, but not IFITM1, in virus-producing HEK293T cells inhibits Env processing and incorporation into virions. We conclude that negative imprinting of HIV-1 infectivity does not occur through disruption of Env clustering and may occur through alternative mechanisms, perhaps involving the modification of the viral membrane. In addition, IFITM3 may reduce infectivity by interfering with Env processing and incorporation into virions in a cell type-specific manner.

## 2. Materials and Methods

### 2.1. Cell Lines, Plasmids, and Reagents

Human embryonic kidney HEK293T/17 cells were purchased from ATCC (Manassas, VA). HeLa-derived TZM-bl cells (donated by Drs. J.C. Kappes and X. Wu [49]) were received from the NIH HIV Reagent Program. The cells were cultured in high-glucose Dulbecco’s Modified Eagle Medium (DMEM, Corning, NY, USA) supplemented with 10% heat-inactivated fetal bovine serum (FBS, Atlanta Biologicals, Flowery Branch, GA, USA) and 100 units/mL penicillin/streptomycin (Gemini Bio-Products, West Sacramento, CA, USA). The HEK293T/17 cells growth medium was supplemented with 0.5 mg/mL of Geneticin (Life Technologies, Inc., Grand Island, NY, USA).

The pCAGGS plasmid expressing the IFITM-sensitive HXB2 Env and pSVIII vector expressing the IFITM-resistant AD8 Env were kindly provided by Dr. J. Binley (Torrey Pines Institute, San Diego CA, USA) and Dr. J. Sodroski (Dana-Farber Cancer Institute, Boston, MA, USA), respectively. The GFP-Vpr plasmid was a gift from Dr. T. Hope (Northwestern University, Evanston, IL, USA). The pR9ΔEnvΔNef HIV-1-based packaging vector and pcRev, have been described previously [34]. pQCXIP vector-based constructs encoding human IFITM1 and IFITM3 were a gift from Dr. A.L. Brass [24].

The viral protease inhibitor, Saquinavir (SQV), human HIV-1 immunoglobulin (HIV IG), anti-p24 capture antibody 183-H12-5C (CA183), Chessie 8 mouse mAb for gp41 (produced from HIV-1 gp41 Hybridoma cells (Cat#526)), and monoclonal antibody (2G12) to HIV-1 gp120 were obtained from the NIH HIV Reagent Program. Other antibodies used were rabbit recombinant antibody for IFITM3 (Abcam, Waltham, MA, USA, cat. no. ab109429), rabbit anti-IFITM1 (Sigma, St. Louis, MO, USA, cat. no. HPA004810), goat anti-HIV gp120 (Fitzgerald, Crossville, TN, USA, cat. no. 20HG-81), mouse anti-GAPDH (Proteintech, Rosemont, IL, USA, cat. no. 60004), goat anti-human IgG HRP (ThermoFisher Scientific, Waltham, MA, USA, cat. no. 31412), mouse anti-rabbit IgG HRP (Millipore, Temecula, CA, USA, cat. no. AP188P), rabbit anti-mouse IgG HRP (EMD Millipore, St. Louis, MO, USA, cat. No. AP160P), donkey anti-goat IgG HRP (Santa Cruz Biotechnology, Dallas, TX, USA, cat. no. sc-2020), and goat anti-human IgG (H+L) conjugated with AlexaFluor 647 (AF647, ThermoFisher, cat. No. A21445). A 16% paraformaldehyde stock was purchased from ThermoFisher (cat. No. 28906). The DMEM without phenol red was obtained from Life Technologies.

### 2.2. Pseudovirus Production and Characterization

HIV-1 pseudoviruses were produced by transfecting HEK293T/17 cells using JetPRIME transfection reagent (Polyplus-transfection, SA, New York, NY, USA), as described previously [34]. HEK293T/17 cells grown in 6-well tissue culture plates were transfected with HXB2 Env or AD8 Env-expressing plasmid (0.6 μg), pR9ΔEnvΔNef (0.8 μg), eGFP-Vpr (0.14 μg), pcRev (0.2 μg), or 0.3–1.0 μg of either pQCXIP-IFITM1, pQCXIP-IFITM3, or empty pQCXIP expression vectors. To produce immature viruses, 300 nM of SQV was added to a growth medium, as indicated. The transfection medium was replaced with phenol-free DMEM/10%FBS after 10–12 h, and the cells were further incubated for an additional 34–36 h at 37 °C, 5% CO_2_ incubator, after which, the virus-containing culture medium was collected, passed through a 0.45 μm filter, and concentrated 10× using Lenti-X concentrator (Clontech, Mountain View, CA, USA). The cells were harvested and lysed, as described elsewhere [50]. The virus was precipitated after an overnight concentration with Lenti-X by centrifuging at 1500× *g* for 45 min at 4 °C, resuspended in PBS, and kept at −80 °C. The p24 content of pseudoviruses and protein content of cell lysates were determined by ELISA [51] and BCA [50] assays, respectively.

### 2.3. Western Blotting

p24 normalized pseudovirus samples or whole-cell lysates were loaded onto 4–15% polyacrylamide gel (Bio-Rad, Hercules, CA, USA) and blotted onto 0.45 μm nitrocellulose membrane (Cytiva, Marlborough, MA, USA). The membranes were blocked with 10% dry milk in 0.1% Tween 20 in phosphate-buffered saline solution. The membranes were incubated overnight at 4 °C with different primary antibodies as follows: HIV IG (1:2000 dilution), rabbit anti-IFITM1 (1:500 dilution), a recombinant anti-fragilis antibody for IFITM3 (1:1000), mouse anti-GAPDH (1:2000), and mouse anti-HIV gp41 (1:10). Secondary antibody staining was performed using either goat anti-human HRP, mouse anti-rabbit HRP, or rabbit anti-mouse HRP for 1 h at room temperature using a dilution of 1:3000. The chemiluminescence signal was recorded on ChemiDoc XRS+ (Bio-Rad) using Image Lab version 5.2 software and analyzed using Image Lab 5.2 software.

### 2.4. Infectivity Assay

To determine the infectivity of HIV-1 pseudoviruses, TZM-bl cells were seeded into black clear-bottom 96-well plates. The cells were infected with serially diluted pseudoviruses and centrifuged at 4 °C for 30 min at 1550× *g* to aid virus attachment to cells. The infectivity was measured 48 h post-infection by lysing the samples with the Bright-Glo luciferase substrate (Promega, Fitchburg, WI, USA) at room temperature and immediately reading the luciferase signal using a TopCount NXT reader (PerkinElmer Life Sciences, Shelton, CT, USA). Specific infectivity was obtained by normalizing to the p24 content and plotted as percentage of specific infectivity of viruses produced by cells transfected with an empty vector. The infectivity results were analyzed using an unpaired Student’s *t*-test implemented in GraphPad Prism version 9.3.1 for Windows (GraphPad Software, La Jolla, CA, USA).

### 2.5. Single Virus-Based IFITM and Env Incorporation Analysis

Eight-chambered glass coverslips (#1.5, Lab-Tek, Nalge Nunc International, Penfield, NY, USA) were coated with the 0.1 mg/mL poly-D-lysine (Sigma, St. Louis, MO, USA). Pseudoviruses diluted 30-fold in PBS^++^ were attached to coverslips for 30 min at room temperature. After washing, the attached pseudoviruses were fixed using 4% paraformaldehyde (PFA) solution in PBS^++^ for 30 min at room temperature. Excess paraformaldehyde was quenched by washing samples with 20 mM Tris in PBS^++^. The pseudoviruses were permeabilized using 0.2% Triton-X100 in PBS^++^ for 10 min at room temperature. The samples were washed thrice with PBS^++^ and blocked with 15% FBS in PBS^++^ for 2 h at room temperature. The samples were then incubated overnight with the primary monoclonal antibody to HIV-1 gp120 2G12 (5 μg/mL), rabbit polyclonal IFITM1 (1:50) or recombinant rabbit IFITM3 (1:100) antibodies at 4 °C. The samples were washed 9 times with 15% FBS and incubated with the secondary goat anti-human AlexaFluor-647 (2 μg/mL) and goat anti-rabbit AlexaFluor-568 (2 μg/mL) antibodies for 1 h at room temperature. The samples were washed 9 times with PBS^++^ before imaging using a wide-field DeltaVision Elite microscope.

### 2.6. Immunostaining and Sample Preparation for dSTORM

Eight-chambered glass coverslips (#1.5, Lab-Tek, Nalge Nunc International, Penfield, NY, USA) were coated with the 0.1 mg/mL poly-D-lysine (Sigma, St. Louis, MO, USA) and 100 nm gold nanoparticles (Cytodiagnostics, G-100-20) according to the previously described protocol [35]. The pseudoviruses diluted in PBS^++^ (30-fold) were attached to pre-treated coverslips for 30 min at room temperature. Unbound pseudoviruses were removed by washing and attached pseudoviruses were fixed using 4% paraformaldehyde solution in PBS^++^ for 30 min at room temperature in the dark. Excess paraformaldehyde was quenched by washing samples with 20 mM Tris in PBS^++^. The samples were blocked with 15% FBS in PBS^++^ for 2 h at room temperature. The FBS solution was removed, and the samples were incubated overnight with primary monoclonal antibody to HIV-1 gp120 (2G12, 5 μg/mL) at 4 °C. The samples were washed 9 times with PBS^++^ and were further used for immunofluorescence imaging and dSTORM experiments. Virus aggregation had no significant impact on our analysis, as evidenced by a very weak correlation (using Pearson coefficient) between Env and GFP-Vpr signals for all four pseudovirus preparations.

### 2.7. Wide-Field Fluorescence and dSTORM Imaging

Initially, the immunostained pseudoviruses were imaged using a wide-field microscope (Elite DeltaVision (Leica Microsystems, Inc., Deerfield, IL, USA)). The UPlanFluo 40×/1.3 NA oil objective (Olympus, Tokyo, Japan) and DAPI/FITC/TRITC/Cy5 Quad cube filter sets (Chroma, Bellows Falls, VT, USA) were used for the imaging.

For dSTORM imaging, the stock solutions of Tris-NaCl buffer, an oxygen scavenging system (GLOX), catalase, and methyl ethylamine were prepared using the Nikon STORM sample preparation protocol [52]. These stock solutions were kept at 4 °C and used within 1–2 weeks of preparation. For experiments, the above-mentioned buffers were immediately mixed in a 90:9:1 ratio to avoid buffer acidification (for each sample every 2 h), as described previously [53]. The imaging buffer was added to the sample, and the glass coverslips were sealed with parafilm to limit the oxygen exchange. dSTORM imaging was performed on an Oxford ONI super-resolution microscope (Nanoimager, San Diego, CA, USA). The imaging was carried out using a 638 nm laser with a 50 ms frame rate, for a total of 20 k frames. The power of the 405 nm laser was adjusted to 0.2 mW to increase the blinking rate of AF647. SMLs with a photon count of more than 1000 and a localization precision lower than 20 nm were considered for analysis.

Image drift correction was performed using the previously published protocol [35] with at least three immobilized fiducial gold nanoparticles in the imaging field. GFP-Vpr particles with fewer than 20 AF647 SMLs were excluded from analysis to eliminate the background signal.

### 2.8. DBCAN Analysis

Coverslip-adhered pseudovirions were identified based on the GFP-Vpr fluorescence signal. The coordinates of the single-molecule localizations (SMLs) were assigned to a virus, using a search radius of 200 nm from the center of a GFP spot. Clustering analysis was performed using density-based spatial clustering of applications with noise (DBSCAN), as described previously [54], using a custom MATLAB script. This algorithm identifies SML clusters based on two user-selectable parameters: the search radius (R) and the minimum number of SMLs (N) within that radius [55,56]. The R of 15 nm appears optimal for selecting smaller and denser clusters at higher SML thresholds [35]. For our analysis of dSTORM images, the threshold was set to R = 15 nm.

### 2.9. Data Processing and Statistical Analyses

For analyzing the immunostaining data, a custom MATLAB script was used to identify GFP-Vpr-labeled pseudoviruses by finding local maxima with a fast 2D peak finder. A specific signal-to-background ratio was fixed to eliminate the faint signals. The coordinates of GFP-Vpr and AF647 were used to quantify the Env fluorescence signals associated with the particles.

The R software (version 4.3.1) was used for the statistical analysis of categorized clustering data (1, 2, >2 clusters or no clusters) using Fisher’s exact test. Distributions of single-molecule localizations per virion measured by 2D dSTORM were analyzed using a custom MATLAB script for a two-sample Kolmogorov–Smirnov (KS) test. To alleviate the effect of a very large sample size for SML data, the results were binned using an optimal bin width set to represent each sample population [54]. Where indicated, to emulate the effect of sample size reduction, we applied optimal binning to the SML data (n > 100 points) sets using the following equation: W (bin width) = 2 (3rd quantile − 1st quantile) × N^−1/3^. The pairwise distance analysis of SMLs on single pseudoviruses was carried out using the *pdist*. MATLAB function. The statistics were processed in MATLAB using a two-sample Kolmogorov–Smirnov (KS) non-parametric test.

Finally, two-way repeated measures beta regression models were employed to determine whether Env clustering differs between the viruses in a panel (e.g., control vs. SQV), for virions with different numbers of clusters (i.e., no clusters, 1, 2, 2+ clusters), and over three biological replicates. These regression models included virus treatment and the number of clusters as main effects, along with their two-way interaction. These regression models were calculated in the full dataset, stratified by SML thresholds (≥20, ≥60, ≥90), and *p*-values are presented unadjusted. As outcome proportions in our data were often inclusive of 0%, and beta regression requires outcome values to be greater than 0% and less than 100%, we added a transformation, described by Smithson and Verkuilen [57], to our outcome proportions prior to any regression analysis. Beta regression analysis was carried out in SAS v.9.4 (SAS Institute, Cary, NC, USA), and statistical significance was evaluated at the 0.05 threshold.

## 3. Results

### 3.1. IFITM1 and IFITM3 Incorporation into HIV-1 Pseudoviruses Inhibits Infectivity but Only IFITM3 Interferes with Processing and Incorporation of Sensitive Envs

To assess the effect of virus-incorporated IFITMs on Env distribution on pseudovirions, we prepared three independent panels of GFP-Vpr-labeled pseudoviruses containing the IFITM-sensitive HXB2 Env [33] or the resistant AD8 Env [32], as described in Methods. Each panel included four viral preparations produced in control HEK293T cells, as well as in cells expressing IFITM1 or IFITM3. Immature pseudoviruses produced in the presence of the HIV-1 protease inhibitor, saquinavir (SQV), were included as a control for virus maturation-driven Env clustering reported previously [35,46,58]. The infectivity of pseudovirus panels was tested on HeLa-derived TZM-bl cells, ectopically expressing CD4 and CCR5, along with endogenous levels of CXCR4 [59]. IFITM1 and IFITM3 expression in producer cells strongly reduced the infectivity of progeny HXB2 pseudoviruses (Figure 1D), in good agreement with the previous studies [11,12,33,43]. We found that maturation (Gag processing) of pseudoviruses produced by IFITM1 or IFITM3 expressing cells was not altered compared to the control, while Gag cleavage was blocked by SQV, as expected (Figure 1A and Appendix A). Both IFITMs were efficiently incorporated into pseudoviruses, as evidenced by prominent IFITM1 and IFITM3 bands on immunoblots of virus and cell lysates (Figure 1C and Appendix A). We also produced a pseudovirus panel with increasing amounts of IFITM1 (0.3–0.75 μg) plasmids. IFITM1 incorporation into pseudovirions increased with the plasmid amount (Appendix A) in line with the previous study [11]. To avoid over incorporating IFITMs into virions, we used the lower amount of plasmids (0.3 μg) which caused a robust reduction in infectivity (Figure 1D).

We next tested whether the reduced infectivity of IFITM-containing pseudoviruses was caused by diminished processing or incorporation of the sensitive HIV-1 HXB2 Env into viral particles. IFITM3, but not IFITM1, expression in virus-producing cells significantly lowered the proteolytic processing and virus incorporation of the gp160 precursor (Figure 1B and Appendix A), in good agreement with the previous studies [18,32]. Inhibition of gp160 processing by IFITM3 was observed across four independent panels of viruses, while reduction in Env cleavage in IFITM1 pseudoviruses did not reach statistical significance (Figure 1E). IFITM3 expression also tended to diminish HIV-1 Env incorporation into pseudoviruses, as measured by immunoblotting (Figure 1B and Appendix A). However, the decrease in HIV-1 Env incorporation in IFITM3-containing pseudoviruses across multiple panels was not statistically significant (Figure 1F). These results imply that the reduced Env processing and somewhat diminished incorporation into IFITM3-containing pseudoviruses are due to the effects of this protein on virus producing cells and not through direct competition with Env incorporation into viral particles.

To further assess the effect of IFITMs on HXB2 Env incorporation on a single particle level, we performed immunofluorescence staining of GFP-Vpr-labeled pseudovirions. Pseudoviruses adhered to glass coverslips were fixed and incubated with a human anti-Env antibody (2G12), followed by staining with a secondary antibody conjugated with AlexaFluor-647 (AF647) (Appendix A). All virus preparations contained comparable levels of GFP-Vpr (Figure 1G and Appendix A), whereas analysis of single virus immunofluorescence revealed a weaker Env signal for IFITM3-containing particles compared to control and IFITM1 pseudoviruses (Figure 1H and Appendix A). The less efficient HXB2 Env incorporation into IFITM3-containing pseudoviruses was observed across four independent virus preparations (Figure 1I), in general agreement with our immunoblotting results.

We next tested the effect of IFITM1 and IFITM3 on resistant AD8 Env pseudoviruses. As expected, AD8 Env was less sensitive to inhibition by IFITM1 and IFITM3 compared with HXB2 Env (Figure 2D), while the IFITMs incorporated equally efficiently into both pseudoviruses (Figure 2C). Maturation of AD8 pseudoviruses was not affected by IFITMs (Figure 2A) [30,32], in line with previously published studies.

Immunoblotting of virus and cell lysates for resistant AD8 Env-containing pseudovirions shows no impact on gp160 processing by IFITMs (Figure 2B) in both virus and cell lysates. No change in Env processing (Figure 2E) or incorporation (Figure 2F) by IFITMs was observed across two independent panels of viruses.

Immunofluorescence staining of GFP-Vpr-labeled pseudovirions showed similar GFP-Vpr incorporation for all four pseudovirus preparations in a panel (Figure 2G). In contrast to sensitive HXB2 Env-containing pseudoviruses, analysis of single virus immunofluorescence revealed a comparable AD8 Env signal for all virus preparations including IFITM3-containing particles (Figure 2H) across two independent virus panels, in agreement with our immunoblotting results. Thus, the IFITMs did not noticeably affect AD8 Env processing or incorporation into virions [30,32].

To further correlate the amount of Env glycoprotein and IFITM incorporation on a single virus level, we fixed and permeabilized GFP-Vpr-labeled pseudoviruses with TX-100 to immunostain for IFITMs using antibodies against the cytoplasmic/intraviral IFITM epitopes. Non-permeabilized viruses were used as a negative control for immunostaining and as a control for the potential disruptive effect of membrane permeabilization with TX-100 (Appendix A). HXB2 and AD8 Env pseudovirions were co-stained for Env. The GFP-Vpr (Appendix A) and Env signals (Appendix A) for permeabilized pseudoviruses were comparable for both sensitive and resistant Env. Reduced Env glycoprotein signals were observed for permeabilized sensitive HXB2 Env pseudovirions consistent with non-permeabilized virions (Appendix A). No change in Env signal was observed for pseudoviruses containing the resistant AD8 Env (Appendix A). We also tested if IFITM incorporation affected HXB2 Env incorporation on a single particle level by correlating IFITM and HXB2 Env signals. The lack of correlation between IFITM and Env incorporation (Appendix A) implies that IFITM incorporation neither competes with nor promotes Env incorporation into pseudoviruses.

### 3.2. IFITM Incorporation Does Not Consistently Perturb Env Clustering on HIV-1 Particles

We next visualized the distribution of Env on HIV-1 GFP-Vpr-labeled pseudoviruses by direct stochastic optical reconstruction microscopy (dSTORM). Single-molecule Env localizations (in red) overlaid onto diffraction-limited images of GFP-Vpr (gray) (Figure 3A) reveal a non-uniform Env distribution on virions, with a tendency to form clusters. Consistent with the reduced Env signals observed in immunoblots and by wide-field fluorescence imaging of single virions, fewer SMLs were detected by dSTORM on IFITM3 pseudoviruses for three independent biological replicates (Figure 3B–D). Lower median SMLs were detected for IFITM3 pseudoviruses compared to other pseudoviruses in independent panels (Figure 3E). To ensure that a statistically significant reduction in the Env SMLs for IFITM3-containing particles was not the result of the large sample size, we applied the optimal binning protocol to reduce the effective sample size of our data [54]. This approach confirmed that pseudoviruses produced by IFITM3- but not IFITM1-expressing cells had, on average, fewer SMLs than control particles (Appendix A).

Env clustering on pseudoviruses was analyzed using the DBSCAN algorithm, which defines SML clusters based upon a user-selected minimal number of SMLs within a search radius [35,55]. For DBSCAN analysis, we kept the search radius constant (R = 15 nm), while varying the SMLs threshold (N) between 20 and 90. From DBSCAN analysis of our dSTORM data, the virions were classified into four categories: virions with no clusters, one cluster, two clusters, and more than two clusters. The relative fractions of virions in each category were plotted as a function of SML thresholds (Figure 4). The more stringent SML thresholding resulted in a lower fraction of virions with Env clusters. This analysis revealed that immature particles contained a greater number of multiple clusters relative to control virions for all three viral preparations (Figure 4A–C), in agreement with previous studies using stimulated emission depletion (STED) microscopy and dSTORM [35,46]. On average, a smaller fraction of IFITM3-containing pseudoviruses exhibited Env clusters compared to control samples (Figure 4A–C). However, the IFITM effects on Env clustering varied between independent preparations and within preparations as a function of the SML thresholds, so the IFITM3 effect on Env clustering was not statistically significant across pseudovirus panels and DBSCAN parameters. The inconsistent effect of IFITM incorporation is suggestive of the lack of robust disruption of Env clusters by these proteins.

To simplify our clustering analysis, we also considered just two categories of viruses—those with and without clusters (regardless of the number of Env clusters). Similar to the four-category analysis above, the fraction of viruses with Env clusters diminished for the more stringent SML thresholds (Appendix A). Here too, varied degrees of Env cluster disruption by IFITMs were observed for three independent pseudovirus panels—from no effect across the SML thresholds to significant inhibition of Env clustering (Appendix A). We also examined the effects of IFITMs on the pairwise distance distributions between all Env SMLs within each single particle. Two out of three preparations showed overlapping distributions of the Env–Env distances for the entire pseudovirus panel, while significantly shorter pairwise Env SML distances for IFITM1- and IFITM3-containing particles compared to the control were observed for one pseudovirus preparation (Appendix A).

To determine whether IFITMs have significant effects on Env clustering across independent virus preparations, we analyzed the pooled results of all three viral panels. The four-category (Figure 5A) and two-category (Figure 5B) analyses of pseudoviruses show no difference in Env clustering in the presence of IFITM1 or IFITM3.

Thus, IFITM incorporation does not appear to disrupt Env clustering on mature pseudoviruses under our experimental conditions.

## 4. Discussion

Our dSTORM data support the HIV-1 maturation-driven coalescence of multiple Env clusters into largely a single focus per virion as reported previously [35,42,46]. A larger fraction of SQV-pretreated immature particles contained multiple Env clusters compared to control pseudoviruses (Figure 4 and Figure 5). In contrast, analysis of independent virus panels revealed that incorporation of IFITMs does not significantly alter Env clustering. Of note, while not reaching statistical significance, IFITM1 and IFITM3 tended to exert opposite effects on the Env distribution on pseudoviruses. A smaller fraction of IFITM3 pseudoviruses contained Env clusters for certain SML thresholds (Figure 3 and Figure 4). The apparent difference in effects of IFITM1 and IFITM3 Env clustering may be explained by fewer SMLs associated with IFITM3 particles compared to control and IFITM1 pseudoviruses. The lower localization density should diminish the apparent Env clustering, as defined by DBSCAN analysis.

It is generally assumed that all three IFITMs act through similar mechanisms and that the spectrum of affected viruses is generally determined by the IFITMs’ subcellular localization in target cells. It appears that IFITMs alter the properties of cell membranes to disfavor the transition from hemifusion to full fusion [17,22]. The mechanism of inhibition of infectivity by HIV-1-incorporated IFITMs is less well defined. Given the lack of significant IFITM effect on Env clustering upon incorporation into HIV-1 pseudoviruses observed in our dSTORM experiments, it appears that IFITMs interfere with Env-mediated fusion via alternative mechanisms, including the modification of the viral membrane. Recent results showing the ability of the IFITM’s amphipathic helix and the conserved intracellular loop to directly bind cholesterol [60,61,62] are in line with the disruption of cholesterol-rich lipid domains. Future studies will be aimed at elucidating the effects of IFITMs on the properties of viral lipid membranes and how these changes modulate the ability of Env glycoproteins to mediate virus-cell fusion.

Our results also revealed that expression of IFITM3, but not IFITM1, in virus producing HEK293T cells can inhibit HIV-1 fusion by interfering with Env processing and incorporation (although not significantly) into progeny pseudoviruses, in agreement with the previous report [18]. This effect of IFITM3 is upstream of virus budding. Note that reduced Env incorporation was not observed in the IFITM3-expressing CD4+ T-cell line [11], and it remains to be determined if this effect occurs in physiologically relevant primary CD4+ T-cells and macrophages. It is also worth pointing out that the extent of HIV-1 Env incorporation into pseudoviruses does not correlate with IFITM3-mediated restriction [11], similar to the effect of IFITM1 on HXB2 pseudovirus infectivity (Figure 1D). However, the effect of IFITM3 on gp160 cleavage in virus-producing cells and, as a result, the reduced gp41/gp160 ratio in HXB2 pseudoviruses (Figure 1B) may be an additional determinant of their reduced infectivity.

## Figures and Tables

**Figure 1 viruses-15-02390-f001:**
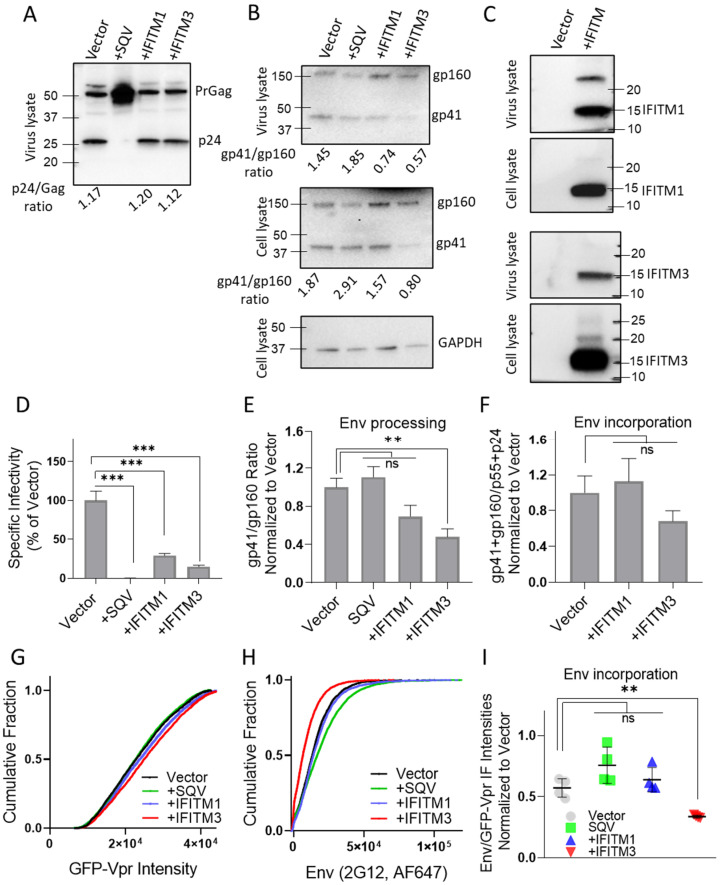
**IFITM incorporation restricts infectivity of HIV-1 pseudoviruses and affects incorporation and cleavage of sensitive Envs.** A panel of four pseudoviruses—Vector (control), +SQV (saquinavir treated, immature), IFITM1, and IFITM3-containing particles—were produced in parallel by transfection of HEK293T/17 cells. (**A**) Analysis of virus maturation by p24 immunoblotting. (**B**) Immunoblotting analysis of virus and cell lysates for Env incorporation and processing for one of the three independent pseudovirus panels (see also Appendix A). (**C**) Western blot analysis of IFITM1 and IFITM3 incorporation into virions. (**D**) Inhibition of HIV-1 pseudovirus infectivity by IFITMs. TZM-bl cells were infected for 48 h with the indicated pseudovirions and the resulting luciferase signal was normalized to control (Vector) particles. Data are means and standard deviations of triplicate values from four independent panels of pseudoviruses. (**E**) The average efficiency of Env precursor (gp160) proteolytic processing for four independent preparations measured by calculating the gp41/gp160 band density ratio. (**F**) Env incorporation assessed by calculating the ratio of the total Env signal (gp41 + gp160) over the sum of the p24 and p55 bands, averaged across four pseudoviral preparations. (**G**,**H**) Immunofluorescence analysis of HIV-1 GFP-Vpr (**G**) and Env (**H**) incorporation into single virions using the anti-gp120 2G12 antibody and anti-human AF647-conjugated secondary antibody. (**I**) The average ratio of Env over GFP-Vpr signal for four independent pseudoviral preparations. The statistical analysis was performed using Student’s *t*-test. Significance: n.s., *p* > 0.05; **, 0.01 > *p* > 0.001; ***, *p* < 0.001.

**Figure 2 viruses-15-02390-f002:**
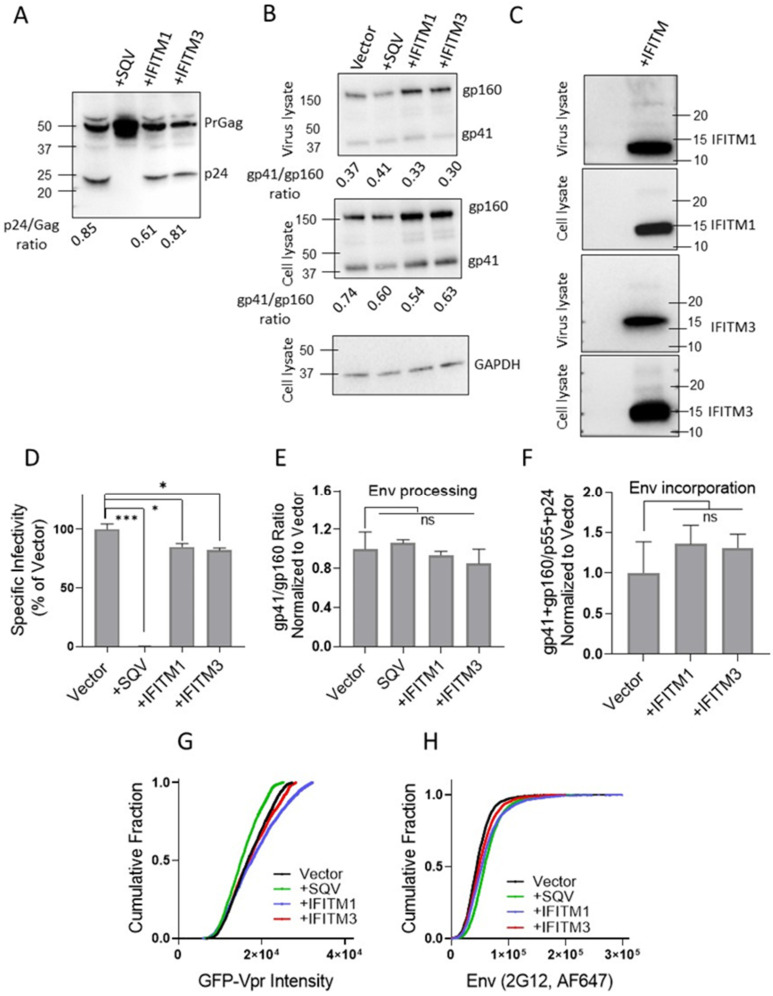
**IFITM incorporation does not affect incorporation and cleavage of resistant HIV-1 AD8 Env**. A panel of resistant AD8 Env pseudoviruses (Vector, SQV-treated (immature), IFITM1, and IFITM3-containing) was produced and analyzed. (**A**) Analysis of virus maturation by p24 immunoblot. (**B**) Immunoblot analysis of Env incorporation and processing for a representative pseudovirus panel. (**C**) Western blot analysis of IFITM incorporation into pseudoviruses. (**D**) Effects of IFITMs on HIV-1 pseudovirus infectivity. TZM-bl cells were infected and the resulting luciferase signal (normalized to Vector) was measured after 48 h. Data are means and standard deviations of triplicate values from two independent panels of pseudoviruses. (**E**) The average efficiency of Env precursor (gp160) proteolytic processing for two independent preparations measured by calculating the gp41/gp160 band density ratio. (**F**) Env incorporation assessed by calculating the ratio of the total Env signal (gp41 + gp160) over the sum of the p24 and p55 bands, averaged across two pseudoviral preparations. (**G**,**H**) Immunofluorescence analysis of HIV-1 GFP-Vpr (**G**) and Env (**H**) incorporation into single virions using the anti-gp120 2G12 antibody and anti-human AF647-conjugated secondary antibodies. The statistical analysis was performed using Student’s *t*-test. Significance: n.s., *p* > 0.05; *, 0.05 > *p* > 0.01; ***, *p* < 0.001.

**Figure 3 viruses-15-02390-f003:**
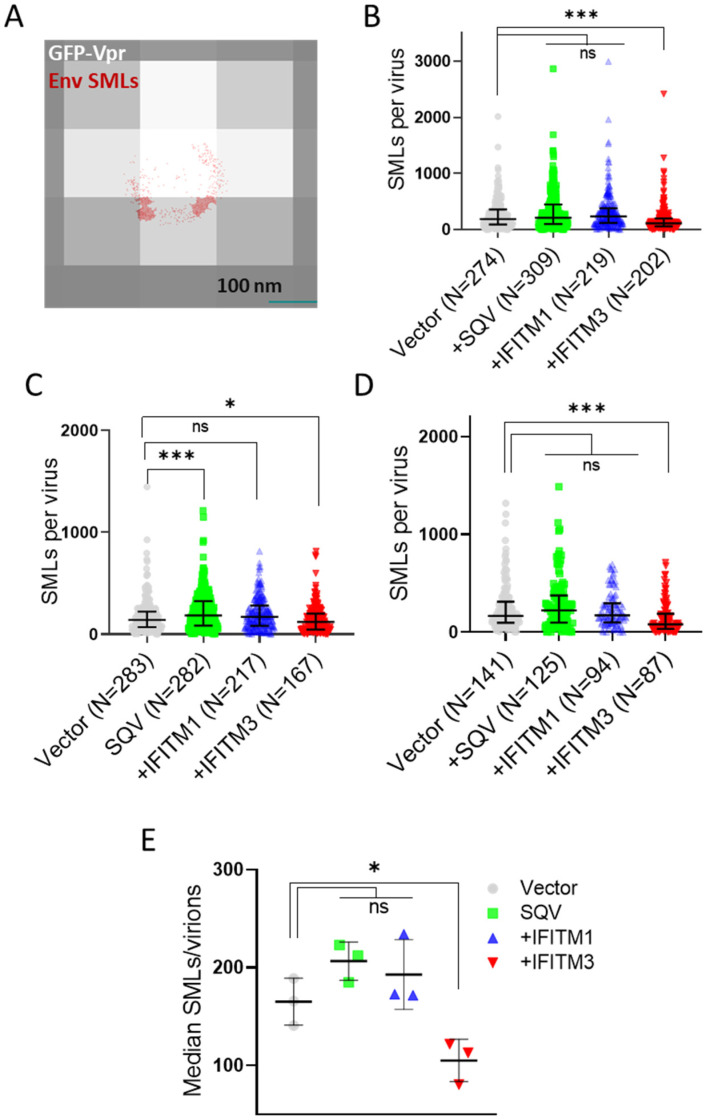
**Single-molecule localization analysis reveals reduced Env incorporation into IFITM3-containing pseudovirions**. (**A**) A representative dSTORM result showing single-molecule localizations (SMLs, red) overlaid onto a diffraction-limited image of a GFP-Vpr-labeled virus (gray). (**B**–**D**) Distributions of single-molecule localizations per virion measured by 2D dSTORM for three independent pseudoviral preparations. Statistical analysis was performed using a custom MATLAB script for a two-sample Kolmogorov–Smirnov (KS) test. (**E**) Median distributions of single-molecule localizations per virion measured by 2D dSTORM for three independent pseudoviral preparations. The statistical analysis was performed using Student’s *t*-test. Significance: n.s., *p* > 0.05; *, 0.05 > *p* > 0.01; ***, *p* < 0.001.

**Figure 4 viruses-15-02390-f004:**
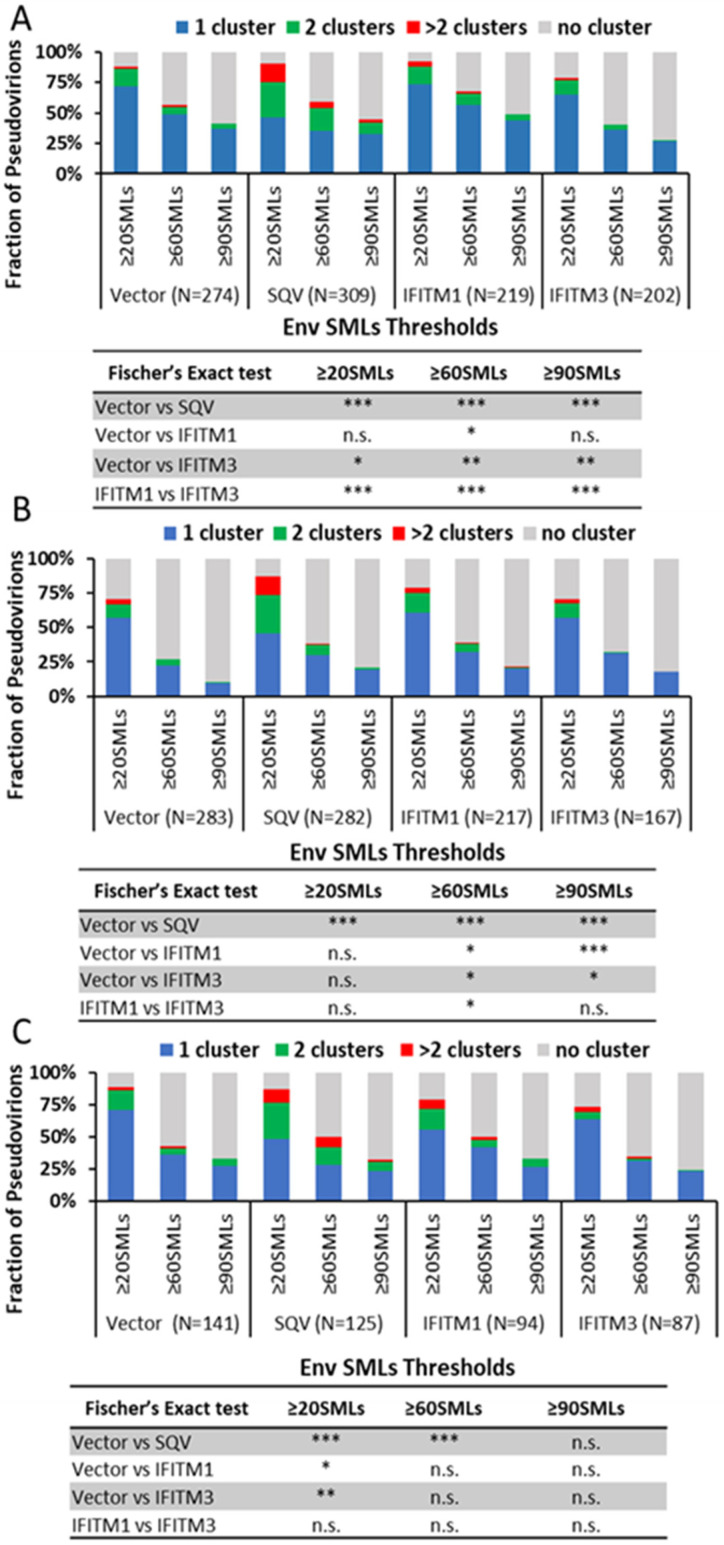
**Effect of IFITMs incorporation on Env distribution on single HIV pseudovirions imaged by 2D dSTORM.** Clusters of Env localizations obtained by dSTORM were defined for three independent pseudoviral preparations by the DBSCAN algorithm for a fixed search radius of 15 nm and varied minimum number of SMLs (from ≥20 to ≥90) within that radius. (**A**–**C**) Fractions of virions containing different numbers of Env clusters: 1, 2, or >2 clusters or no clusters as a function of the DBSCAN single-molecule localization threshold are plotted. The three graphs represent three independent virus preparations. The number of pseudoviruses analyzed (N) is shown in parentheses under the graphs. Statistical analysis was performed using Fisher’s exact test. Significance: n.s., *p* > 0.05; *, 0.05 > *p* > 0.01; **, 0.01 > *p* > 0.001; ***, *p* < 0.001.

**Figure 5 viruses-15-02390-f005:**
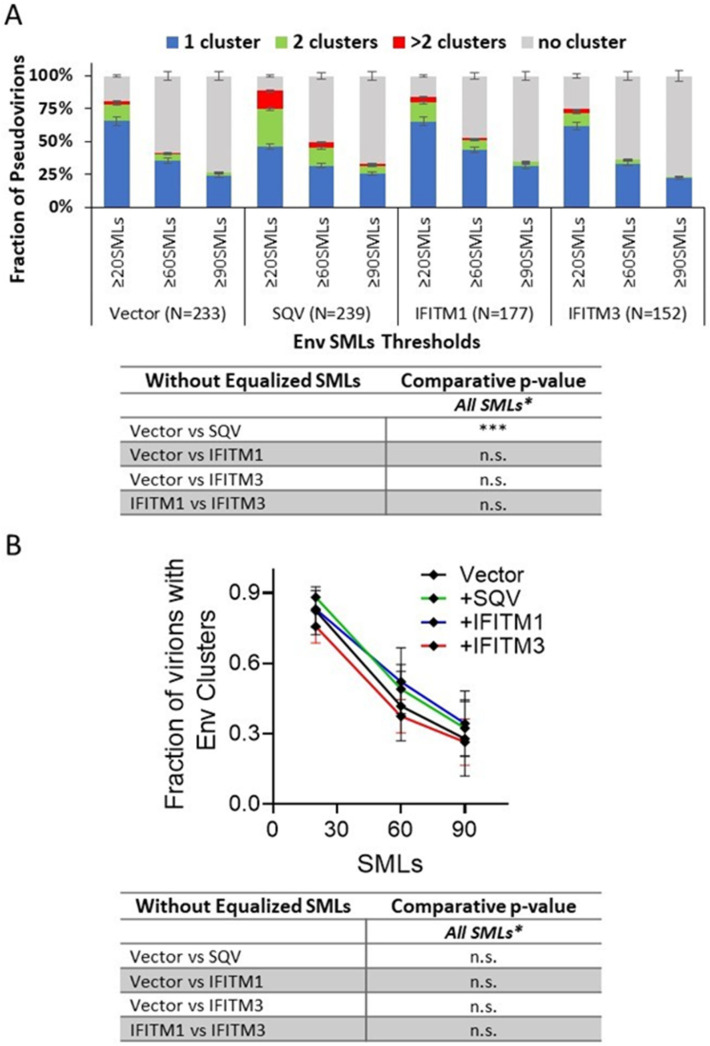
**HIV Env clustering ratio on single viruses in the presence of IFITMs for the average of three independent viral preparations reveals no disruption of Env clusters.** Env clusters were defined by DBSCAN with a 15 nm search radius and varied minimum SMLs within that radius (20, 60, and 90 SMLs). The average fractions of virions containing different numbers of Env clusters are plotted as a function of the DBSCAN single-molecule localization threshold using (**A**) 1, 2, or >2 clusters or no clusters (four-category analysis) and (**B**) pseudoviruses with and without Env clusters (two-category analysis). The average fractions and S.D. for each Env cluster category are plotted in A and B. Statistical analysis is performed using a two-way beta regression model. * Omnibus *p*-values are based on two-way interactions between sample type and number of clusters, assessing if the outcome differences between samples vary by the number of clusters. Significance: n.s., *p* > 0.05; ***, *p* < 0.001.

## Data Availability

All pertinent data is included in the manuscript or available upon request.

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
