# Peer review of "IFITM1 and IFITM3 Proteins Inhibit the Infectivity of Progeny HIV-1 without Disrupting Envelope Glycoprotein Clusters"

_viruses, 2023, doi:10.3390/v15122390_

Round 1

Reviewer 1 Report

Comments and Suggestions for Authors

In this study, Verma and colleagues use dSTORM to experimentally test the hypothesis that IFITM1 and IFITM3 can reduce the infectivity of virion particles produced in their presence, by altering HIV-1 Env clustering. This hypothesis proved correct for SERINC5, antiviral factor that also inhibits membrane fusion.

The dSTORM analyses presented here did not reveal major differences in this step, suggesting therefore that IFITMs block the infectivity of virion particles by other means. Of interest, during their analysis the authors found that virion particles produced in the presence of IFITM3, but not of IFITM1, displayed reduced Env incorporation and Env processing and suggested that this could participate to the infectivity defect driven by IFITM3. Albeit controversially, this specific finding has been reported before, although it remains for the moment unclear whether these Env defects are instrumental to the infectivity defect of IFITM3. Indeed, there is not a simple linear relationship between the number of Env trimers displayed on virions and their infectivity, as this varies greatly among Env proteins of different strains. In this respect, the results presented here could be important to better understand this topic, but a few additions are needed to increase the impact of this study and support the authors conclusion.

Major

1)     This reviewer could not find a WB of Env in cells from which IFITM3 virions were produced. The analysis of Env in these cells is important to show at which point the alterations in processing or levels (perhaps trafficking as well) may occur.

2)     It has been suggested that gp120 could be shed from virion particles. It would therefore be important not to infer this parameter from the gp160/gp41 ratio, but to detect it more directly.

3)     The decrease in Env molecules incorporated into virion particles is of interest but infectivity is influenced by many parameters. Would the authors observe, in their experimental setup, the same drop in Env incorporation also using an Envelope that is resistant to the antiviral effects of IFITM3? This would be an important control to link (or not), infectivity defects caused by IFTIM3 to Env changes.

4)     As the authors mention, the effects of IFITMs on Env clustering did not reach a statistical difference with respect to controls, but a tendency was however observed. Would it be possible to perform this same analysis using a fluorescent IFITM protein (the authors have developed this PMID: 33656312) to correlate directly the levels of IFITMs incorporated into single particles with the behavior on Env clustering ? IFITM incorporation into virions is likely to be heterogeneous and this may flatten differences. The possibility to focus on virions incorporating high levels of IFITM could lead to more conclusive results.

Minor

It is surprising that immature Gag VLPs incorporate just as much Env than mature ones, because in a previous study a defect in Env incorporation was clearly measured (PMID: 14694135). Have the authors an explanation for these differences?

Line 376 ,correct typo: IFITM3 may also slightly promotes

Author Response

 Major

1)     This reviewer could not find a WB of Env in cells from which IFITM3 virions were produced. The analysis of Env in these cells is important to show at which point the alterations in processing or levels (perhaps trafficking as well) may occur.

Author’s Response:

Thank you for the suggestion. The Western blots (for IFITM and Env) for both viral and cell lysates are now included in Figure 1. As expected, based on previous publications, cell lysates show reduced gp160 processing in IFITM3-expressing cells. This suggests that poor gp160 processing and incorporation into IFITM3-containing virions reflect the IFITM3 effects on Env processing and trafficking in virus-producing cells.

2)     It has been suggested that gp120 could be shed from virion particles. It would therefore be important not to infer this parameter from the gp160/gp41 ratio, but to detect it more directly.

Author’s Response:

We did not perform WB experiments to test for gp120, since has been reported previously. However, our IF and STORM single-molecule localization experiments using the anti-gp120 2G12 antibody do not reveal a major loss of gp120 signal in IFITM3-containing virions beyond the signal reduction that is proportional to reduced Env incorporation. This does not suggest a considerable effect of IFITM on gp120 shedding.

3)     The decrease in Env molecules incorporated into virion particles is of interest but infectivity is influenced by many parameters. Would the authors observe, in their experimental setup, the same drop in Env incorporation also using an Envelope that is resistant to the antiviral effects of IFITM3? This would be an important control to link (or not), infectivity defects caused by IFTIM3 to Env changes.

Author’s Response:

We produced and tested IFITM-carrying pseudovirions with both sensitive (HXB2) and resistant (AD8) HIV-1 Env. The results for AD8 pseudoviruses are shown in new Figure 2 and lines 313-328. As expected, AD8 Env was less sensitive to inhibition by IFITM1 and IFITM3 compared with HXB2 Env (Fig. 2D), despite the efficient incorporation of IFITMs into these pseudoviruses (Fig. 2C, H). IFITM3 incorporation into resistant virions did not affect Env processing (Fig. 2E) or incorporation (Fig. 2F).

4)     As the authors mention, the effects of IFITMs on Env clustering did not reach a statistical difference with respect to controls, but a tendency was however observed. Would it be possible to perform this same analysis using a fluorescent IFITM protein (the authors have developed this PMID: 33656312) to correlate directly the levels of IFITMs incorporated into single particles with the behavior on Env clustering? IFITM incorporation into virions is likely to be heterogeneous and this may flatten differences. The possibility to focus on virions incorporating high levels of IFITM could lead to more conclusive results.

Author’s Response:

We produced and tested RFP-Vpr-based pseudoviruses containing the functional IFITM3-mNeonGreen construct (see below). However, the NeonGreen fluorescence associated with single RFP-Vpr labeled pseudoviruses was very weak, suggesting that single virus imaging is not sufficiently sensitive to reliably detect virus-incorporated IFITM3-mNeonGreen. In contrast, Western blotting detected virus-incorporated fluorescent IFITM3. Another caveat of this single virus fluorescence-based approach is that a similar fluorescently tagged IFITM1 construct lacks antiviral activity (unpublished results). For this reason, we adopted an immunolabeling approach (Figs. S4 and S5) to correlate IFITM incorporation with Env incorporation on a single-particle basis.

Minor 

It is surprising that immature Gag VLPs incorporate just as much Env than mature ones because, in a previous study, a defect in Env incorporation was clearly measured (PMID: 14694135). Have the authors an explanation for these differences?

Author’s Response:

As the Murakami et al. paper and other studies have shown, the infectivity defect in immature HIV-1 is due to the impairment of Env function, but not incorporation of Env into virions. This makes sense, since maturation (Gag/Pol cleavage) occurs after virus assembly and budding from cells.

Line 376, correct typo: IFITM3 may also slightly promote

Author’s Response: Corrected.

Reviewer 2 Report

Comments and Suggestions for Authors

This study focuses on the role of IFITM 1 and IFITM3 in modulating Env clustering in HIV-1. They found that although these proteins affect Env processing, they do not have a role in modulating Env clustering. The study reports a single observation, however, the results are of importance to the HIV-1 research community.

There are some concerns that need to be addressed before this study can be published.

Major points

1)      What is the expression level of IFITM 1 and 3 in the producer cells? The authors tranfected 300 ng of IFITM 1 or IFITM 3 in producer cells. Did the authors assess whether the incorporation of IFITM into virus particles has reached saturation?

2)      The authors show that viruses with IFITM3 have reduced Env processing. Was there a dose dependency between the amount of IFITM incorporated into virus particles and Env processing? Also, it would be interesting to see if one increases the amount of IFITM1 incorporation in virus particles by transfecting more IFITM 1 in producer cells, does the difference become significant?

3)      Figure 1G: Please provide representative immunofluorescence images in the supplementary figure

4)      Line 274: for data not shown, please provide the data in the supplementary file

5)      For the immunofluorescence analysis, the authors can also stain for IFITM 1 and IFITM3 and get an idea of the no. of IFITM1 and 3 containing virus particles. The lesser effect on Env incorporation and env processing in IFITM 1 containing viruses might be merely due to the fact that there are fewer viral particles containing IFITM1 than IFITM 3

6)      Figure 2 B-D and Figure S2 seem to be replicas. Please explain.

7)      Line 383: inhibits both Env processing and incorporation: Inhibition of Env incorporation by IFITM3 is not significant (Figure 1F). Please re-write the line.

Minor points

1)      Figure 1A: Please mention how the specific infectivity was calculated in the figure legend.

2)      The authors have mentioned IFITM1 containing viruses exhibiting more clusters compared to control only in the discussion section. Please include this observation in the results section too.

3)      Please move Figure 3 to supplementary. The pooled results for the figure are shown in Figure 4A. So the three independent analyses can be moved to supplementary.

4)      Title: The study does not dive deep into the mechanism of action of IFITM 1 vs. 3 in HIV-1. It shows that both IFITM 1 and 3 do not affect Env clustering in HIV-1. Therefore I would suggest changing the title of the manuscript so that it clearly aligns with the observation of the study. 

Author Response

 Major points

1)      What is the expression level of IFITM 1 and 3 in the producer cells? The authors transfected 300 ng of IFITM 1 or IFITM 3 in producer cells. Did the authors assess whether the incorporation of IFITM into virus particles has reached saturation?

Author’s Response:

We thank the reviewer for this suggestion. The new Figures 1 and 2 now show the expression levels of IFITMs in producer cells for both sensitive (HXB2, Figure 1) and resistant (AD8, Figure 2) Env-carrying pseudovirions. Additionally, we produced pseudovirions using increasing amounts of IFITM plasmids (0.3-1 mg). IFITM1 and IFITM3 incorporation into pseudovirions increased with the plasmid amount but did not reach saturation under these conditions (Figs. S2B and S2C). Since the lowest amount of IFITM plasmids quite significantly reduced pseudovirus’ specific infectivity, we opted not to use higher amounts of plasmids, in order to avoid overexpression and over-incorporation of these proteins.

2)      The authors show that viruses with IFITM3 have reduced Env processing. Was there a dose dependency between the amount of IFITM incorporated into virus particles and Env processing? Also, it would be interesting to see if one increases the amount of IFITM1 incorporation in virus particles by transfecting more IFITM 1 in producer cells, does the difference become significant?

Author’s Response: According to the densitometric analysis, as we increase the IFITM1 plasmid from 0.3 to 0.75 ug the Env cleavage increases for IFITM1 incorporated virions (Fig. S1D).

3)      Figure 1G: Please provide representative immunofluorescence images in the supplementary figure

Author’s Response: We added a representative image of single pseudoviruses containing GFP-Vpr (green) and immunostained for Env glycoprotein (red) using the anti-gp120 2G12 antibody and AF647-conjugated secondary antibody (Fig. S3).

4)      Line 274: for data not shown, please provide the data in the supplementary file-

Author’s Response: Immunostaining and GFP-Vpr intensity distribution data are now included for all the pseudovirus preparations. Please refer to Figures 1G and Figs. S1D, S1I.

5)      For the immunofluorescence analysis, the authors can also stain for IFITM 1 and IFITM3 and get an idea of the no. of IFITM1 and 3 containing virus particles. The lesser effect on Env incorporation and env processing in IFITM 1 containing viruses might be merely due to the fact that there are fewer viral particles containing IFITM1 than IFITM 3

Author’s Response: To correlate the amount of Env glycoprotein (incorporation and processing) and IFITM incorporation in single viruses, we fixed and permeabilized GFP-Vpr labeled pseudoviruses to expose the intraviral IFITM epitopes to antibodies. Viruses were stained for IFITMs and Env (sensitive (HXB2) or resistant (AD8)). Non-permeabilized viruses were used as controls (Fig. S4). Please refer to Supplementary Figures S4 and S5 for details. The GFP-Vpr and Env glycoprotein signals for permeabilized and non-permeabilized pseudoviruses were comparable, while only a very low, background level staining for IFITMs for non-permeabilized particles. Since anti-IFITM1 and IFITM3 antibodies are different, a direct comparison of IFITM1 and IFITM3 incorporation is not possible. An additional paragraph is added to describe these results in lines 347-361.

6)      Figure 2 B-D and Figure S2 seem to be replicas. Please explain.

Author’s Response: These figures show single-molecule localization by 2D STORM revealing reduced Env incorporation into IFITM3-containing pseudovirions. However, Fig. 2B-D shows the KS test without optimal binning of data, while new Fig. S7 (previous Fig. S2) shows the KS test after the implementation of the optimal binning technique for all SMLs.

7)      Line 383: inhibits both Env processing and incorporation: Inhibition of Env incorporation by IFITM3 is not significant (Figure 1F). Please re-write the line.

Author’s Response: Please refer to lines 271-277 in the main manuscript. The revised text is as follows: “Inhibition of gp160 processing by IFITM3 was observed across four independent panels of viruses, while reduction of Env cleavage in IFITM1 pseudoviruses did not reach statistical significance (Fig. 1E). IFITM3 expression also tended to diminish HIV-1 Env incorporation into pseudoviruses, as measured by immunoblotting (Fig. 1B and Figs. S1C, S1H). However, the decrease in HIV-1 Env incorporation in IFITM3-containing pseudoviruses across multiple panels was not statistically significant (Fig. 1F).”

 Minor points

1)      Figure 1A: Please mention how the specific infectivity was calculated in the figure legend.

Author’s Response: We preferred putting the details of specific infectivity calculations in the Materials and Methods section. Please refer to Materials and Method subsection 2.4 infectivity assay for details.

2)      The authors have mentioned IFITM1 containing viruses exhibiting more clusters compared to control only in the discussion section. Please include this observation in the results section too.

Author’s Response: Since the very modest enhancing effect was not significant, we deleted this sentence to avoid confusion.

3)      Please move Figure 3 to supplementary. The pooled results for the figure are shown in Figure 4A. So the three independent analyses can be moved to supplementary.

Author’s Response: We agree with the reviewer but prefer to show the results for a representative preparation as the main figure.

4)      Title: The study does not dive deep into the mechanism of action of IFITM 1 vs. 3 in HIV-1. It shows that both IFITM 1 and 3 do not affect Env clustering in HIV-1. Therefore, I would suggest changing the title of the manuscript so that it clearly aligns with the observation of the study.

Author’s Response: Thank you for the suggestion. The title of the manuscript has been modified to “IFITM1 and IFITM3 Proteins Inhibit Infectivity of Progeny HIV-1 without Disrupting Env Clusters.”

Reviewer 3 Report

Comments and Suggestions for Authors

In this manuscript, Verma and co-workers tried determining the mechanism by which IFITM1 and 3 restrict HIV-1. Their main conclusion is that both restriction factors used different mechanisms to decrease viral infectivity. The experimental work is sound and extremely well done, and the article is clear, concise, and easy to read. The strength of this article is the careful methodological approach to test if these restriction factors alter Env clustering. While their data cannot prove or disprove the initial hypothesis, their findings are extremely interesting and relevant.

The only weakness in this article is the cell line used, which is not extremely relevant to a real infection. Therefore, it will be desirable if they expand the discussion to contextualize more relevant cell lines and results obtained from others about the relative expression of the restriction factors in these cell lines.

Finally, I would like to thank the authors for taking such care to describe the experimental section.

Author Response

Author’s Response: We are grateful to the reviewer for the very positive comments.

Reviewer 4 Report

Comments and Suggestions for Authors

Verma et al. provide an analysis of whether IFITM1 or IFITM3 impact HIV-1 Envelope clustering inside virions. This study involves impressive technical prowess in order to study a very important phenomenon in the HIV research community. However, the authors provide merely a negative result and do not provide any new information on the antiviral properties of IFITM proteins. It is possible that the authors are simply unable to detect a meaningful difference according to how they perform the study, which is of course a limitation to the interpretation of the data. I appreciate the distinctions observed between impacts of IFITM1 and IFITM3 in their experiments. However, the results presented here do not necessarily translate to what IFITM1 and IFITM3 may be doing against HIV in CD4+ T cells or macrophages. The authors should include more cautionary language and should cite additional literature to provide a greater context and to allow for more extensive discussion of what their findings mean.

Major:

1.     It was previously shown by Compton et al. CHM 2014 that IFITM3 inhibits cell-cell spread and virion infectivity to a much greater extent than IFITM1. Can the authors provide more transparency about how their specific infectivity results were performed and quantified?

2.     Along the same lines, IFITM3 does not inhibit HIV-1 Env incorporation when viruses are produced in CD4+ T cells (Compton et al CHM 2014). These results should be discussed and the authors should state that their own results (using virus produced in HEK293T) do not necessarily explain what may or what may not be happening in CD4+ T cells, macrophages, or other tissues infected by HIV-1 in vivo.

3.     Furthermore, Ahi et al. mBio 2020 demonstrated that, while human IFITM3 can decrease MLV Env incorporation into MLV virions, its impact on HIV Env incorporation into HIV virions is much more modest. Furthermore, using different amounts of Env during the pseudotyping process, the authors showed that IFITM3 must inhibit retrovirus infectivity in a manner that cannot be solely explained by a deficiency in Env incorporation into virions. That means that IFITM3 is inhibiting Env function, not just its incorporation into virions. This is reinforced by the fact that MLV glycoGag can rescue infectivity in IFITM3+ virions in a manner that does not involve restoring Env incorporation levels. Therefore, glycoGag restores Env function, not quantity, in IFITM3+ virions. Hence the rationale for investigating whether IFITMs affect Env clustering, as the authors checked in this current submission. However, their lack of significant findings should not be the basis for concluding that IFITMs do not inhibit Env function. The authors would need to follow the lead of Ahi et al. in order to exclude the role of differential Env incorporation in infectivity inhibition by IFITM3. So, it would be very interesting to reassess the impact of IFITM3 on Env clustering in virus preparations that contain the SAME amount of incorporated Env (compare IFITM3-negative and IFITM3-positive virus that were made to incorporate similar amounts of Env). At a minimum, the authors should discuss this in the text. Of course, it would be better if they launched additional experiments to address this.

4.     The authors have previously shown that IFITM3 inhibits the infectivity of pseudoviruses bearing HA. Does IFITM3 reduce HA incorporation in this context? If not, can the authors discuss how they believe infectivity is inhibited in these pseudovirions?

Author Response

Major:

  1. It was previously shown by Compton et al. CHM 2014 that IFITM3 inhibits cell-cell spread and virion infectivity to a much greater extent than IFITM1. Can the authors provide more transparency about how their specific infectivity results were performed and quantified?

Author’s response: We used pR9dEnvΔNef backbone to produce pseudovirions and performed our infectivity assay on HeLa-derived TZM-bl cells. The 10X concentrated pseudovirions by Lenti-X were serially diluted 10x in triplicates to infect the 40%-45% confluent TZM-bl cells in 96-well plates. The resulting luciferase signal was measured at 46-48 hours post-infection. Specific infectivity was obtained by normalizing to the p24 content of the viral preparation and plotted as a percentage of specific infectivity of viruses produced by cells transfected with an empty vector. For further details, please refer to Materials and Methods (subsection 2.4).

  1. Along the same lines, IFITM3 does not inhibit HIV-1 Env incorporation when viruses are produced in CD4+ T cells (Compton et al CHM 2014). These results should be discussed and the authors should state that their own results (using virus produced in HEK293T) do not necessarily explain what may or what may not be happening in CD4+ T cells, macrophages, or other tissues infected by HIV-1 in vivo.

Author’s response: We thank the reviewer for this comment and revised the Results and Discussion accordingly. Please refer to the Discussion section, lines 480-488.

  1. Furthermore, Ahi et al. mBio 2020 demonstrated that, while human IFITM3 can decrease MLV Env incorporation into MLV virions, its impact on HIV Env incorporation into HIV virions is much more modest. Furthermore, using different amounts of Env during the pseudotyping process, the authors showed that IFITM3 must inhibit retrovirus infectivity in a manner that cannot be solely explained by a deficiency in Env incorporation into virions. That means that IFITM3 is inhibiting Env function, not just its incorporation into virions. This is reinforced by the fact that MLV glycoGag can rescue infectivity in IFITM3+ virions in a manner that does not involve restoring Env incorporation levels. Therefore, glycoGag restores Env function, not quantity, in IFITM3+ virions. Hence the rationale for investigating whether IFITMs affect Env clustering, as the authors checked in this current submission. However, their lack of significant findings should not be the basis for concluding that IFITMs do not inhibit Env function. The authors would need to follow the lead of Ahi et al. in order to exclude the role of differential Env incorporation in infectivity inhibition by IFITM3. So, it would be very interesting to reassess the impact of IFITM3 on Env clustering in virus preparations that contain the SAME amount of incorporated Env (compare IFITM3-negative and IFITM3-positive virus that were made to incorporate similar amounts of Env). At a minimum, the authors should discuss this in the text. Of course, it would be better if they launched additional experiments to address this.

Author’s response: We agree and apologize for misleading the reviewer into thinking that we believe that the only mode by which IFITM3 inhibits HIV-1 infectivity is through reduction of Env processing or incorporation. This was not our intention. It is clear, however, that, unlike IFITM1, IFITM3 is inhibiting Env processing/incorporation, at least in HEK293T cells. We revised the Discussion to clarify this point to avoid confusion. As to additional experiments, we have now included the results for IFITM-resistant AD8 Env which exhibits a modest reduction in infectivity and unaltered Env processing/incorporation. The inhibitory effect of IFITM3 on AD8 infectivity in the context of equal Env incorporation appears to be modest. We also added a discussion of the results by Ahi et al. in Discussion.

  1. The authors have previously shown that IFITM3 inhibits the infectivity of pseudoviruses bearing HA. Does IFITM3 reduce HA incorporation in this context? If not, can the authors discuss how they believe infectivity is inhibited in these pseudovirions?

Author’s response: That is an interesting question. We have not systematically examined the effect of IFITMs on HA incorporation into pseudoviruses, but plan to perform these experiments in the future. Till then, we would not like to speculate regarding the mechanism of IFITM restriction of IAV. Of note, a recent paper reported the reduction of HA incorporation into VLPs and bona fide IAV (https://doi.org/10.3929/ethz-b-000482464) by IFITM3, so we may see the same effect.  

Round 2

Reviewer 1 Report

Comments and Suggestions for Authors

The authors have addressed this reviewer's comments

Reviewer 2 Report

Comments and Suggestions for Authors

The authors have addressed all my concerns. 

Reviewer 4 Report

Comments and Suggestions for Authors

The authors have addressed all of my comments/concerns.